# Local and Global Spectral Features for Hyperspectral Image Classification

Zeyu Xu [1], Cheng Su [1,2,*], Shirou Wang [1] and Xiaocan Zhang [1,2]

1   School of Earth Science, Zhejiang University, Hangzhou 310030, China
2   Key Laboratory of Geoscience Big Data and Deep Resource of Zhejiang Province, School of Earth Sciences, Zhejiang University, Hangzhou 310030, China
*   Correspondence: sc20184@zju.edu.cn

**Abstract:** Hyperspectral images (HSI) contain powerful spectral characterization capabilities and are widely used especially for classification applications. However, the rich spectrum contained in HSI also increases the difficulty of extracting useful information, which makes the feature extraction method significant as it enables effective expression and utilization of the spectrum. Traditional HSI feature extraction methods design spectral features manually, which is likely to be limited by the complex spectral information within HSI. Recently, data-driven methods, especially the use of convolutional neural networks (CNNs), have shown great improvements in performance when processing image data owing to their powerful automatic feature learning and extraction abilities and are also widely used for HSI feature extraction and classification. The CNN extracts features based on the convolution operation. Nevertheless, the local perception of the convolution operation makes CNN focus on the local spectral features (LSF) and weakens the description of features between long-distance spectral ranges, which will be referred to as global spectral features (GSF) in this study. LSF and GSF describe the spectral features from two different perspectives and are both essential for determining the spectrum. Thus, in this study, a local-global spectral feature (LGSF) extraction and optimization method is proposed to jointly consider the LSF and GSF for HSI classification. To increase the relationship between spectra and the possibility to obtain features with more forms, we first transformed the 1D spectral vector into a 2D spectral image. Based on the spectral image, the local spectral feature extraction module (LSFEM) and the global spectral feature extraction module (GSFEM) are proposed to automatically extract the LGSF. The loss function for spectral feature optimization is proposed to optimize the LGSF and obtain improved class separability inspired by contrastive learning. We further enhanced the LGSF by introducing spatial relation and designed a CNN constructed using dilated convolution for classification. The proposed method was evaluated on four widely used HSI datasets, and the results highlighted its comprehensive utilization of spectral information as well as its effectiveness in HSI classification.

**Keywords:** convolutional neural network (CNN); global spectral feature; hyperspectral image classification (HSIC); local spectral feature

## 1. Introduction

Hyperspectral images (HSI) generally contain hundreds of subdivided spectral bands captured at a continuous wavelength [1]. Compared with multispectral images, HSI significantly increases the spectral dimension while retaining the spatial dimension. With rich spectral information, HSI enables the detection, identification, and discrimination of target materials at a more detailed level [2] and is widely used in geology [3,4], agriculture [5,6], environmental studies [7,8], quantitative inversion [9,10] and other fields [11]. Due to its powerful spectral characterization ability, HSI has also served as one of the most significant data sources in the remote sensing community, especially for some classification or identification applications, such as Land Use and Land Cover (LULC) mapping [12,13]. This rich spectral information brings unique advantages to HSI, but it also presents challenges for image processing. On the one hand, the HSI band number and data dimension have a

geometric multiple increase compared with those of traditional multispectral images, which leads to a significant increase in the calculation as well as the curse of dimensionality [14]. The increased data volume also requires more samples to support, and the acquisition of refined labels is difficult [15]. On the other hand, the adjacent bands of remote sensing images are more strongly correlated to closer bands. The interval between bands in HSI is significantly reduced, resulting in strong redundancy and band correlation [16], which increases the difficulty of mining hidden information from HSI. Therefore, for HSI classification, it is vital to identify a method that can effectively use the rich spectral information.

Spectral information is typically represented by a spectral curve in an HSI. Thus, a large number of single curve analysis techniques are proposed to describe the spectral characteristics, and most of them contain the idea of dimensionality reduction. (1) Some methods use band selection to pick up the characteristics of the original spectral curve with a series of standards (e.g., spectral distance [17], spectral divergence [18] and spectral variance [19]) for subsequent processing. Demir et al. [20] used feature-weighting algorithms to obtain weights for all bands and selected bands with higher weights for classification. (2) Feature extraction methods describe and calculate the curve features, such as shape and variation, by designing a series of indices for classification (e.g., the spectral angle mapper (SAM)). Chen et al. [21] combined the SAM feature and maximum likelihood classification (MLC) with the magnitude and shape features for classification. They discovered that adding SAM could improve accuracy. He et al. [22] proposed a handcrafted feature extraction method based on multiscale covariance maps. The spectral features were obtained by computing the covariance matrices of the central pixels at various scales, where the values were the covariance of the spectral band pairs. They found that using the multiscale covariance maps as input features could greatly improve the classification accuracy. (3) In order to suppress the redundant information and highlight the useful information for a better description of the spectral curve, spectral transformation methods convert the original spectral space to another feature space via mathematical transformations. The most typical spectral transformation method is the principal component analysis (PCA). Jiang et al. [23] proposed a superpixel-wise PCA (SuperPCA) approach that considers the diversity in different homogeneous regions and was able to incorporate spatial context information. Although SuperPCA is an unsupervised method, its performance is comparable to supervised approaches. Fu et al. [24] proposed a PCA and segmented PCA (SPCA)-based multiscale 2-D-singular spectrum analysis (2-D-SSA) fusion method for joint spectral–spatial HSI feature extraction and classification. The method can extract multiscale spectral–spatial features and outperform other state-of-the-art feature-extraction methods. However, the above spectral information utilization methods describe the spectral characteristics based on manual design, which are generally limited in type and quantity, and it is also difficult to extract deeper and more representative information [25]. Thus, these methods have limitations in the face of the complex spectral information in HSI, and they are difficult to fully use the spectral information to describe the characteristics of the targets. In recent years, breakthroughs have been made in artificial intelligence. Data-driven deep learning methods can automatically learn features at different levels of the data for classification and have achieved remarkable success in the field of computer vision. As a typical deep learning model, convolutional neuronal network (CNN) has also been widely used to analyze the spectral curve [26] and has been shown to have the potential to surpass traditional methods that utilize manually designed features [27,28]. The 1D CNN uses 1D convolution kernels to extract features on the single spectral curve and combines them through a deep network structure. Hu et al. [29] built a 1D CNN consisting of five layers with weights for HSI classification. The experimental results demonstrated the effectiveness of the proposed method when compared with traditional methods such as SVM.

However, the linearly arranged single spectral curve limited the expression of spectral relationship because the spectral information cannot be effectively aggregated in this structure. In order to utilize more information, there are some methods that treat the spectral information as non-curve data for feature extraction and classification. Some

methods consider the spectral information as a cube. The 2D CNN methods use 2D convolution kernels on a 3D HSI cube for feature extraction, mainly for better use of spatial information. Kussul et al. [30] compared the performances of 1D CNNs and 2D CNNs for land cover and crop classification. They showed that 2D CNNs outperformed 1D CNNs, although some small objects in the classification maps provided by 2D CNNs were smoothed and misclassified. Song et al. [31] proposed a deep 2D CNN based on residual learning and fused the outputs of different hierarchical layers. Their proposed network can extract deeper features and achieve state-of-the-art performance. The 3D CNN methods use 3D convolution kernels on a 3D HSI cube for feature extraction to fuse both spatial and spectral information effectively. Hamida et al. [32] introduced a 3D CNN for jointly processing spectral and spatial features, as well as establishing low-cost imaging. They also proposed a set of 3D CNN schemes and evaluated their feasibility. Li et al. [33] proposed a novel 3D CNN that takes full advantage of both spectral and spatial features. In addition, there are also some 3D CNNs mixed with other convolutional kernels to complement their advantages [34,35]. These CNN methods use a convolution operation to extract spectral features. However, the convolution operation focuses on extracting the features of adjacent data owing to their local perception characteristics. As a result, these methods mainly extract the local spectral features (LSF) for classification. The LSF reflect the local statistical information of the spectrum and describe the relationship between adjacent bands in a local neighborhood (i.e., the local change rate of the original curve). The advantage of LSF is that they can effectively express the characteristics of spectral changes as the wavelength gradually increases and reduces the noise interference. However, the spectral curve is expressed sequentially in a one-dimensional manner, and the distance between bands increases linearly with the increase in wavelength, which leads to the possibility that the distance between effective bands (features) may be large. Thus, the disadvantage of LSF is that they are limited to processing the long-distance spectral relationship, resulting in insufficient expression of complete spectral information. To address this issue, there are some methods that consider the spectral information as an image. Yuan et al. [36] reshaped the 1D spectral vector into a 2D spectral image. In the 2D spectral image, the long-distance spectra from the original 1D spectral vector can be aligned closely or even directly connected, which significantly expands the spectral coverage of the feature extraction window and is conducive to obtaining more spectral feature patterns. This method is conducive to constructing another feature, which is opposite to LSF and describes the relationship between non-adjacent bands in a long-distance span (i.e., the global shape of the original curve), named global spectral features (GSF). The advantage of GSF is that they can selectively construct spectral relationships between arbitrary bands to reflect the characteristics of targets (e.g., the Normalized Difference Water Index uses the long-distance spectral relationship of the green and near-infrared bands to describe the characteristics of water). However, it is challenging to identify a suitable band combination for GSF extraction among a large number of bands in an HSI.

In general, the LSF and GSF describe the spectral information at local and global levels, respectively, and are the embodiment of the characteristics of the targets in different aspects. Thus, for the feature extraction method, the LSF and GSF should be fully considered to effectively express and utilize spectral information. Considering the importance of LSF and GSF when utilizing spectral information, a local-global spectral feature (LGSF) extraction and optimization method is proposed in this study to effectively combine both LSF and GSF for HSI classification. First, we analyzed the limitations of the traditional spectral feature extraction strategy that uses the 1D spectral vector as input, and to obtain more diverse spectral features, the 1D spectral vector was transformed into a 2D spectral image for feature extraction. Second, to extract the LSF, convolution was used as a local feature descriptor to aggregate adjacent spectral statistical information. Third, to extract the GSF, all spectral bands were combined in pairs and modeled automatically by the fully connected layers to introduce the distance-independent GSF upon the LSF, and further fused to form the LGSF. Fourth, to increase the effectiveness of LGSF in classification, LGSF was

optimized by maximizing the difference between classes and increasing feature separability. Fifth, based on the LGSF of each pixel, a dilated convolution-based network with multiple receptive fields was designed for classification, and the pixel class was obtained. Moreover, we noticed the importance of spatial information for HSI classification and enhanced the LGSF with spatial relation to comprehensively utilize spectral and spatial information. To demonstrate the efficiency of the proposed method, we evaluated it on four widely used HSI datasets with several comparison methods. The experimental results demonstrated that the proposed method significantly enhanced the classification accuracy with a more comprehensive use of spectral features.

The contributions of this study are as follows:

- A hyperspectral image classification method combining the local and global spectral features is proposed in this paper.
- We transformed the original spectrum from a 1D spectral curve into a 2D spectral image for feature extraction. The spectral reorganization enhances the spectral connection and is beneficial to obtain more diverse spectral features.
- The image processing and feature extraction methods for the 2D image were used to analyze the spectral information, which could extract more sufficient and stable spectral features with higher class separability.

The rest of this article is organized as follows: In Section 2, the proposed method is introduced in detail. In Section 3, the experiment details are introduced, including the datasets, modeling parameters and comparison methods. In Section 4, the experiment results are analyzed and discussed. In Section 5, the content of the article is concluded.

## 2. Methods

### 2.1. Overview of the Proposed Method

The HSI contains rich spectral information and has a great potential for classifying targets. However, the hidden complex spectral features within HSI increase the difficulty of effectively using them. Although CNN can learn to extract features automatically, its local perception characteristic limits the extracted features to cover only the local spectral range (i.e., LSF) and ignores the long-distance global spectral range (i.e., GSF). Thus, in this study, both the LSF and GSF were extracted, combined, and optimized for HSI classification.

Figure 1 shows an overview of the proposed method. The spectral information of each pixel in the HSI corresponds to a 1D spectral vector or spectral curve. In this study, to increase the correlation between spectra, the original 1D spectral vector of each pixel was transformed into a more compact 2D spectral image for subsequent feature extraction and classification. Next, the local spectral feature extraction module (LSFEM) was designed to extract the LSF from the 2D spectral image, and the global spectral feature extraction module (GSFEM) was designed to extract the GSF from the extracted LSF and to join them to obtain the LGSF. Moreover, the loss function for spectral feature optimization (SFOL) was designed to further optimize the effectiveness of the extracted LGSF automatically, based on the idea of maximizing the separability between classes. Finally, the LGSF of each pixel was input into a classification network built using dilated convolutions to determine the category of the corresponding pixel. To improve the robustness of classification, we also introduced a spatial relation to enhance the LGSF of the pixels to be classified.

The proposed method consists of six parts, and their corresponding sections are as follows: (1) transformation from a 1D spectral vector into a 2D spectral image (Section 2.2); (2) extraction of LSF (Section 2.3); (3) extraction of GSF and combination of LGSF (Section 2.4); (4) optimization of LGSF (Section 2.5); (5) structure of classification network (Section 2.6); (6) spatial enhancement of the LGSF (Section 2.7).

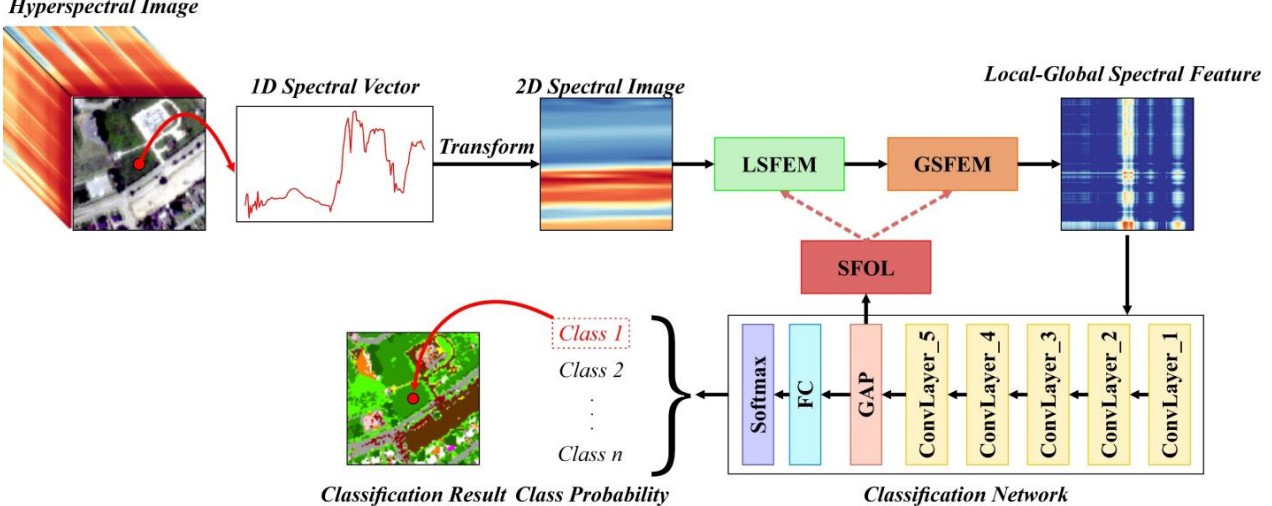

**Figure 1.** The overview of the proposed method.

## 2.2. Transformation of Spectrum from 1D to 2D

To enhance the spectral connection and obtain more complex and diverse spectral features, in this study, the spectral information represented by 1D form in HSI was first converted into 2D. The details of the transformation and feature comparison between 1D and 2D are as follows.

Traditional spectral feature extraction is generally based on a 1D spectral vector. However, in such a sequential 1D structure, the distance between bands increases linearly with the increase in wavelength, which leads to a long distance between the small and large wavelength bands. Because convolution is a local feature extractor, the long-distance relationships are difficult to capture based on the 1D structure. To solve this problem, we transformed the 1D spectral vector of the center pixel in the HSI patch into a 2D spectral image (shown by the black dotted lines in Figure 2) and used it as the basic spectral input data for the subsequent feature extraction. In the 2D spectral image, each band can be adjacent to more bands, which can be seen as the addition of a large number of shortcuts between long-distance bands in the 1D spectral vector. The green lines in Figure 2 show the feature extraction of the 1D spectral vector and 2D spectral image using the corresponding 1D and 2D convolution kernels. Theoretically, compared with the 1D combination, there are more spectral bands and larger spectral coverage in the 2D combination, which makes it more convenient to obtain complex and diverse spectral features.

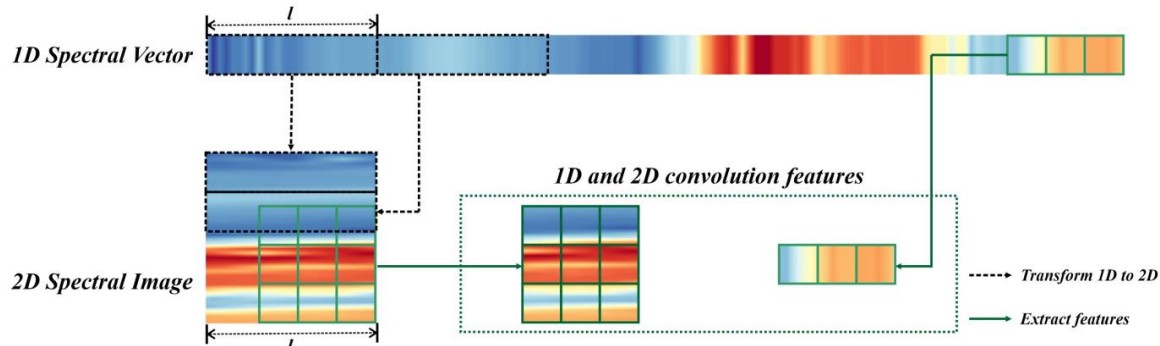

**Figure 2.** The process of transforming 1D spectral vector into 2D spectral image and the comparison of features obtained using 1D and 2D convolution kernels.

Notably, the conversion of 1D spectral vector into the 2D spectral image requires square numbers of bands. If the size of the 1D spectral vector is $1 \times b$, we first interpolate

the original 1D spectral vector to modify the band number to a specific square number, denoted as $l^2$ ($l \in Z$), and then transform it into a 2D spectral image with size $l \times l$.

### 2.3. Local Spectral Feature Extraction Module

LSF is an effective spectral feature that aggregates adjacent spectra and describes the local variation of the spectral curve. To extract the LSF, the LSFEM was proposed and its characteristics are presented below.

Considering that the convolution operation is a widely used local feature extractor that meets the requirements of LSF extraction, in this study, the LSFEM utilizes convolutions to extract LSF, as shown in Figure 3.

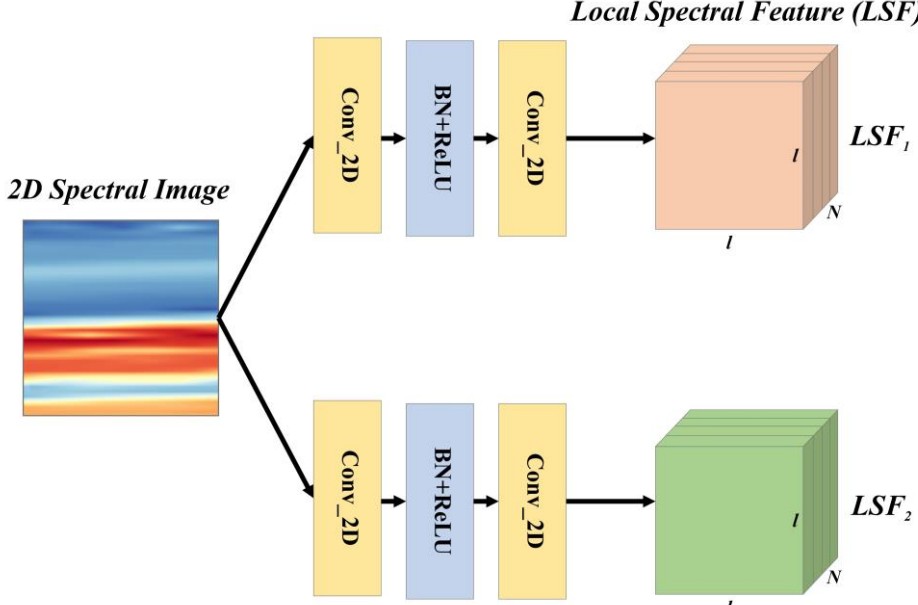

**Figure 3.** The architecture of the LSFEM. The input of the LSFEM is the 2D spectral image and the output of the LSFEM is two LSF groups.

The input of the LSFEM is a 2D spectral image, denoted as $I_{2D}$. Two groups of 2D convolution kernels were used to aggregate the local spectra upon $I_{2D}$ and obtain the LSF. The LSFEM calculation is express in Equation (1):

$$LSF_1 = Conv_2[B(Conv_1(I_{2D}))]$$
$$LSF_2 = Conv_4[B(Conv_3(I_{2D}))]$$

(1)

where $Conv_1$, $Conv_2$, $Conv_3$, and $Conv_4$ are four convolution groups with $N$ kernels; $B$ is the batch normalization and ReLU operation; while $LSF_1$ and $LSF_2$ are two groups of LSF with both sizes of $N \times l \times l$. $N$ is a hyperparameter and was set to 4 in this study.

### 2.4. Global Spectral Feature Extraction Module

GSF is another effective spectral feature that connects bands with spans and describes the relative shape of the spectral curve. To extract the GSF, the GSFEM was proposed and its characteristics are presented below.

Considering the characteristics of GSF, which only depend on the spectral values between bands and are independent of the distance, in the GSFEM, the spectral bands were scattered and combined in pairs. Due to the relationship complexity between bands, in the GSFEM, the data-driven method was used to directly and automatically learn the construction of inter-band features, rather than a manually designed method, as shown in Figure 4.

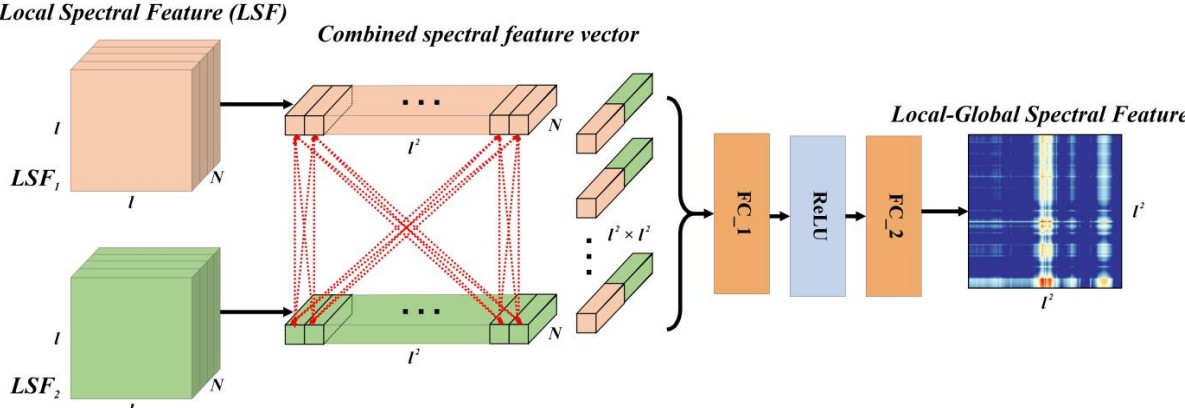

**Figure 4.** GSFEM architecture. The input to the GSFEM is represented by two LSF groups, while the output is the LGSF.

The input of the GSFEM is $LSF_1$ and $LSF_2$ and was obtained using the LSFEM, where each pixel represents an $N$-D feature of one band. Pixels in $LSF_1$ and $LSF_2$ were combined to model GSF. If the pixel of row $i$ and column $j$ in $LSF_1$ is $P_{ij}$, and the pixel of row $i'$ and column $j'$ in $LSF_2$ is $P'_{i'j'}$, the combined spectral feature vector of these two pixels (bands) would be $PP'_{ii'jj'}$, whose size is $2 \times N$. $PP'_{ii'jj'}$ represents an input into a network with two fully connected (FC) layers to compute the GSF of the two target bands. By traversing all combinations of bands in $LSF_1$ and $LSF_2$ and modeling them using the network, a feature map with a size of $l^2 \times l^2$ can be obtained. The generation of this map includes the extraction of LSF and GSF and their fusion, which gives the map the potential to express local as well as global spectral features. Thus, we refer to it as LGSF. Equation (2) shows the modeling process of the pixels of rows $a$ and $b$ in LGSF from $P_{ij}$ and $P'_{i'j'}$ in $LSF_1$ and $LSF_2$, where $\oplus$ is the concatenation operation, $FC_1$ and $FC_2$ are the two FC layers, and $R$ is the ReLU layer. Equation (3) shows the correspondence between subscript parameters.

$$PP'_{iji'j'} = P_{ij} \oplus P'_{i'j'}$$
$$LGSF_{ab} = FC_2[R(FC_1(PP'_{ijxy}))] \tag{2}$$

$$a = i \times l + j$$
$$b = i' \times l + j' \tag{3}$$

### 2.5. The Loss Function for Spectral Feature Optimization

In LSFEM, the LSF is extracted through the convolution operation, whereas in GSFEM, the GSF is further superimposed on the LSF to obtain the final LGSF that contains both local and global feature information. Obviously, the quality of LGSF depends on the feature extraction from the LSFEM and GSFEM, and is controlled by internal learnable parameters. For classification tasks, an effective feature is supposed to have a high inner-class similarity but a low inter-class similarity. To move towards this goal, in this study, inspired by info noise contrastive estimation loss (infoNCE loss) in the contrastive learning domain, we designed the SFOL to constrain the update direction of parameters in LSFEM and GSFEM.

LGSF was considered as the input of a CNN (introduced in Section 2.6) for classification with a batch size of $B$. After a series of convolutions, pooling, and global average pooling layers, a group of feature vectors (FVs) with a size of $B \times K$ was obtained, where $K$ is the channel number of the last convolution layer. The FVs are high-level abstract representations of LGSF and are directly input into a classifier (e.g., the FC layer) to determine the final classification result. Thus, it is necessary to maximize the differences between the classes of the FVs.

Next, if $m$ and $n$ are two training samples in one batch, the category consistency as well as similarity of their FVs were computed. By traversing all sample pairs, a matrix of category consistency ($M_{cc}$) and a matrix of similarity of FVs ($M_{sim}$) can be obtained as described by Equations (4) and (5), whose sizes are both $B \times B$.

$$M_{cc_{mn}} = \left\{ \begin{array}{ll} 1 & C(m) == C(n) \\ 0 & C(n) \mathbin{!}= C(n) \end{array} \right\} \tag{4}$$

$$M_{sim_{mn}} = Norm(FV_m) \otimes Norm(FV_n^T) \tag{5}$$

$C$, $Norm$, and $\otimes$ represent the class of samples, normalized operations, and matrix multiplication, respectively.

Sample pairs with the same category were defined as positive samples, while those with different categories were defined as negative samples. Based on $M_{cc}$ and $M_{sim}$, the similarity between positive samples ($SS_{pos}$) and negative samples ($SS_{neg}$) can be computed using Equations (6) and (7):

$$SS_{pos} = M_{sim} \times (M_{cc} - I) \tag{6}$$

$$SS_{neg} = M_{sim} \times (1 - M_{cc}) \tag{7}$$

where $I$ is the identity matrix used to eliminate the influence of sample pairs composed of the same sample (diagonal elements).

The $SS_{pos}$ was averaged and concatenated with the $SS_{neg}$ to form a new similarity vector (SV). In SV, the first element is the average similarity of all positive samples, and the other elements are the similarities of the negative samples. To increase the similarity between positive samples and reduce the similarity between negative samples, the value of the first element of the SV should approach 1, whereas that of the other elements should approach 0. This is similar to the use of a one-hot code to represent the first class in a classification task. Therefore, we created a pseudo-classification task, whose input is the SV divided by a normalized parameter (i.e., the temperature parameter in the infoNCE loss and was set to 0.07, which is widely used in relevant studies [37]), and label is the one-hot code of the first class, to auxiliary optimize the parameters with the cross-entropy loss. In addition, the length of the SV may change significantly because the number of negative samples is not fixed in a batch. To fix the SV length, the top 20 negative samples with the highest similarity were selected because the negative samples with higher similarity were more likely to confuse the classifier and need more attention.

### 2.6. Dilated Convolution-Based Network

After the processing steps in Sections 2.2–2.5 the spectral vector (spectral curve) of a single pixel is represented by a 2D LGSF image. This LGSF image was further classified to obtain the pixel categories. To better utilize the information hidden in the LGSF, a CNN based on dilated convolution was designed and used for classification, as the dilated convolution can significantly enlarge the receptive field and extract multiscale image features [38].

The overall architecture of the CNN is shown in Figure 5, which takes the LGSF as input and consists of five convolution layers ('ConvLayer' in Figure 5), one global average pooling layer ('GAP' in Figure 5), one fully connected layer ('FC' in Figure 5), and one softmax layer ('Softmax' in Figure 5). Each ConvLayer contains four dilated convolutions ('Dconv' in Figure 5, the numbers in parentheses represent the dilation rate, kernel size and output channel, respectively) with various dilation rates and kernel sizes but the same number of output channels to extract multiscale features. Following batch normalization and ReLU ('BN + ReLU' in Figure 5), these features were concatenated and fused by another dilated convolution. Max pool ('Maxpool' in Figure 5) was used to decrease the feature

size, whose results are the output of the current ConvLayer as well as the input of the next ConvLayer. The features after the five ConvLayers as well as the GAP layer were fed into a FC layer for classification, which contains one linear layer with an input size of 16 and an output size that is the same as the class number. The cross-entropy loss function was used for optimization.

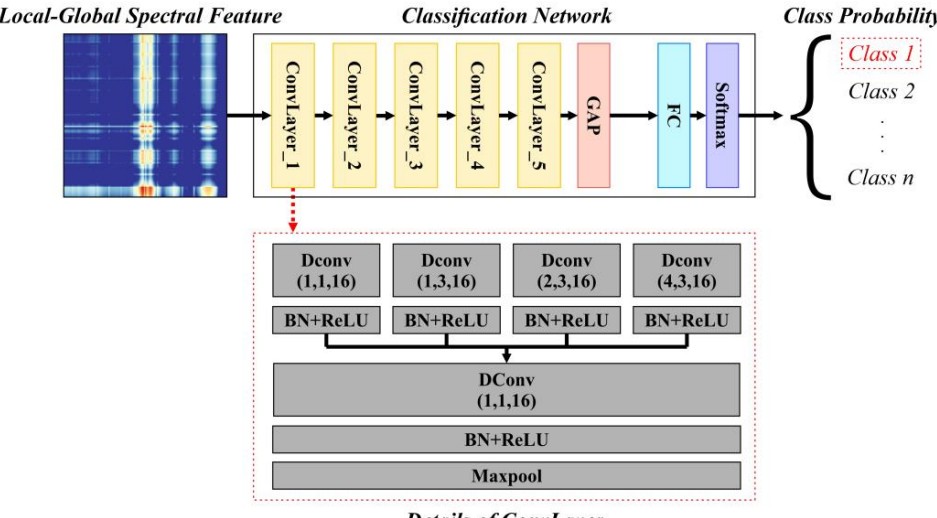

**Figure 5.** The architecture of the classification network. The input of the network is the LGSF, and the output of the network is the category of the pixel that generates the LGSF.

### 2.7. Enhancement of LGSF with Spatial Relation

In the above sections, we introduced the process of extracting the LGSF and using a CNN to obtain the classification results. However, this pattern is applicable only when the input is the spectral vector of a single pixel. Considering the noise interference in HSI, introducing spatial constraints on the LGSF has a significant positive effect on improving robustness.

To enhance the LGSF at the spatial level, we defined a spatial window in the HSI, and the LGSF of each pixel in the spatial window was combined. Suppose that the size of the HSI patch in the spatial window is $H \times W \times N$, where $H$ and $W$ are the height and width of the spatial window, respectively, and $N$ is the band number of the HSI. The LGSF for each pixel in the HSI patch was extracted to a size of $N \times N \times HW$. We used a simple and effective fusion method, namely averaging, to directly combine and fuse the LGSF and obtain the spatially enhanced LGSF with a size of $N \times N \times 1$, as shown in Figure 6. The spatially enhanced LGSF was used as an input feature of the network introduced in Section 2.6 for classification, the result of which represents the class of the center pixel in the HSI patch.

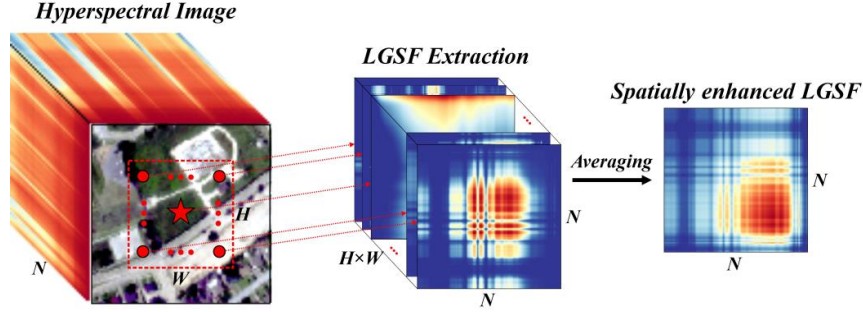

**Figure 6.** The spatial enhancement process of LGSF.

## 3. Experiments

To verify the effectiveness of our proposed method and ensure its universality, we compared it with a series of classical and advanced HSI classification methods using four standard and widely used HSI datasets. Section 3.1 introduces the details of the four HSI datasets, as well as the preparation of the training and test samples. Section 3.2 introduces the training and modeling details as well as the experimental environment. Section 3.3 presents the comparison methods and evaluation metrics.

### 3.1. Datasets

1. Houston 2013

The Houston 2013 dataset was provided by the IEEE Geoscience and Remote Sensing Society (GRSS). The HSI was acquired by the NSF-funded Center for Airborne Laser Mapping (NCALM) over the University of Houston campus and the neighboring urban area, which consists of 144 spectral bands from 380 to 1050 nm with a spatial resolution of 2.5 m and an image size of 349 × 1905. It also contains a total of 15,029 ground truth samples with 15 classes and a pre-defined training and test sample division strategy (details are listed in Table 1), which was also used in this study.

**Table 1.** The number of training and test samples used in the Houston 2013 dataset.

| Class No. | Class Name | Training Samples | Test Samples |
|:---:|:---:|:---:|:---:|
| 1 | Healthy grass | 198 | 1053 |
| 2 | Stressed grass | 190 | 1064 |
| 3 | Synthetic grass | 192 | 505 |
| 4 | Trees | 188 | 1056 |
| 5 | Soil | 186 | 1056 |
| 6 | Water Soil | 182 | 143 |
| 7 | Residential | 196 | 1072 |
| 8 | Commercial | 191 | 1053 |
| 9 | Road | 193 | 1059 |
| 10 | Highway | 191 | 1036 |
| 11 | Railway | 181 | 1054 |
| 12 | Parking Lot 1 | 192 | 1041 |
| 13 | Parking Lot 2 | 184 | 285 |
| 14 | Tennis Court | 181 | 247 |
| 15 | Running Track | 187 | 473 |
| | Total | 2832 | 12,197 |

2. Houston 2018

The Houston 2018 dataset was provided by the 2018 IEEE GRSS Data Fusion Contest and acquired by the National Center for Airborne Laser Mapping over the University of Houston campus and its neighborhood. The HSI covers a 380–1050 nm spectral wavelength range with 48 bands at a 1 m ground sampling distance of size 1202 × 4172. The ground truth samples contained 20 classes and had a higher spatial resolution but smaller spatial extent, with an image size of 1202 × 4768. Thus, the spatial extent and resolution of the HSI and the ground truth sample image were unified using clipping and resampling, resulting in an image size of 601 × 2384 and a total sample number of 504,712. In this study, we randomly selected 100 samples from each class for training and the rest were used for testing (details are listed in Table 2).

3. Pavia University

The Pavia University dataset was acquired using the Reflective Optics System Imaging Spectrometer sensor over Pavia University, Northern Italy. The HSI has a size of 610 × 340, with 103 bands, a spectral wavelength range of 430–860 nm, and a spatial resolution of 1.3 m. The dataset contained 20 classes, with a total number of 42,776 ground truth samples.

In this study, we randomly selected 100 samples from each class for training and the rest were used rest for testing (details are listed in Table 3).

**Table 2.** The number of training and test samples used in the Houston 2018 dataset.

| Class No. | Class Name | Training Samples | Test Samples |
|:---:|:---:|:---:|:---:|
| 1 | Healthy grass | 100 | 9699 |
| 2 | Stressed grass | 100 | 32,402 |
| 3 | Artificial turf | 100 | 584 |
| 4 | Evergreen trees | 100 | 13,488 |
| 5 | Deciduous trees | 100 | 4948 |
| 6 | Bare earth | 100 | 4416 |
| 7 | Water | 100 | 166 |
| 8 | Residential buildings | 100 | 39,662 |
| 9 | Non-res. buildings | 100 | 223,584 |
| 10 | Roads | 100 | 45,710 |
| 11 | Sidewalks | 100 | 33,902 |
| 12 | Crosswalks | 100 | 1416 |
| 13 | Major thoroughfares | 100 | 46,258 |
| 14 | Highways | 100 | 9749 |
| 15 | Railways | 100 | 6837 |
| 16 | Paved parking lots | 100 | 11,375 |
| 17 | Unpaved parking lots | 100 | 49 |
| 18 | Cars | 100 | 6478 |
| 19 | Trains | 100 | 5265 |
| 20 | Stadium seats | 100 | 6724 |
| | Total | 2000 | 502,712 |

**Table 3.** The number of training and test samples used in the Pavia University dataset.

| Class No. | Class Name | Training Samples | Test Samples |
|:---:|:---:|:---:|:---:|
| 1 | Asphalt | 100 | 6531 |
| 2 | Meadows | 100 | 18,549 |
| 3 | Gravel | 100 | 1999 |
| 4 | Trees | 100 | 2964 |
| 5 | Painted metal sheets | 100 | 1245 |
| 6 | Bare Soil | 100 | 4929 |
| 7 | Bitumen | 100 | 1230 |
| 8 | Self-Blocking Bricks | 100 | 3582 |
| 9 | Shadows | 100 | 847 |
| | Total | 900 | 41,876 |

4.　Salinas Valley

The Salinas Valley dataset was collected using the AVIRIS sensor over Salinas Valley, CA, USA. The image has a size of $512 \times 217$ pixels, with 204 bands, a spectral wavelength range of 360–2500 nm, and a spatial resolution of 3.7 m. The dataset contained 16 classes with a total number of 54,129 ground truth samples. In this study, we randomly selected 100 samples from each class for training and the rest were used for testing (details are listed in Table 4).

**Table 4.** The number of training and test samples used in the Salinas Valley dataset.

| Class No. | Class Name | Training Samples | Test Samples |
|:---:|:---:|:---:|:---:|
| 1 | Brocoli_green_weeds_1 | 100 | 1909 |
| 2 | Brocoli_green_weeds_2 | 100 | 3626 |
| 3 | Fallow | 100 | 1876 |
| 4 | Fallow_rough_plow | 100 | 1294 |

**Table 4.** *Cont.*

| Class No. | Class Name | Training Samples | Test Samples |
|:---:|:---:|:---:|:---:|
| 5 | Fallow_smooth | 100 | 2578 |
| 6 | Stubble | 100 | 3859 |
| 7 | Celery | 100 | 3479 |
| 8 | Grapes_untrained | 100 | 11,171 |
| 9 | Soil_vinyard_develop | 100 | 6103 |
| 10 | Corn_senesced_green_weeds | 100 | 3178 |
| 11 | Lettuce_romaine_4wk | 100 | 968 |
| 12 | Lettuce_romaine_5wk | 100 | 1827 |
| 13 | Lettuce_romaine_6wk | 100 | 816 |
| 14 | Lettuce_romaine_7wk | 100 | 970 |
| 15 | Vinyard_untrained | 100 | 7168 |
| 16 | Vinyard_vertical_trellis | 100 | 1707 |
| | Total | 1600 | 52,529 |

### 3.2. Method Modeling

All deep learning methods were trained with the following hyperparameters: a batch size of 32, a learning rate of 0.001 with a decay rate of 0.8 every 20 epochs; an optimizer of stochastic gradient descent (SGD) with a momentum of 0.9, weight decay of 0.0001, and a total training epoch of 500. All models were trained using HSI patches with a spatial size of $11 \times 11$.

All experiments in this study were implemented using Pytorch [39] on a single computer, and the environment was as follows: Windows operating system, Intel (R) Core (TM) i9-10900 K, 64 GB RAM, and GPU of NVIDIA GeForce RTX 3090 with 24 GB GPU memory.

### 3.3. Method Comparison

To verify the effectiveness of our proposed method, we compared it with classical and state-of-the-art HSI classification methods based on machine learning and deep learning. These methods focus on using different features in the HSI, the details of which are described as follows.

SVM [40]: The Support Vector Machine (SVM) is one of the most important machine learning methods widely used for classification. An SVM was implemented using the Python library of Sklearn. We used the 'linear' kernel function and the default values for the other parameters. This is a typical method that directly uses the original spectral curve for classification without feature extraction.

1D CNN [29]: Hu et al., performed pioneering work on HSI classification based on CNN. The proposed 1D CNN contains a 1D convolution layer, max pooling layer, and two-layer FC. The kernel number of the convolution layer was 20. This is a typical method used for extracting LSF for classification.

R2D [36]: Yuan et al., reshaped each pixel of the HSI from 1D into 2D to increase the spectral coverage of convolution and obtain more diverse features. The reshaped 2D images were used as input for classification using the same classification network proposed in this study. This method can be regarded as extracting and using an optimized LSF for classification.

2D CNN [41]: This 2D CNN contains three convolution groups and a two-layer FC. Each convolution group contains two convolution layers and one max-pooling layer. The kernel numbers of the six convolution layers were 32, 32, 64, 64, 128, and 128, the node numbers of the FC were 1024 and the class number. The input HSI was masked by a spectral attention module that can be obtained using global convolution with a nonlinear activation function. This is a typical method that focuses on spatial rather than spectral features for classification.

3D CNN [32]: Hamida et al., proposed and evaluated a set of 3D CNNs, which were widely used in subsequent studies. The 3D CNN contains four 3D convolutions layers, two 3D max pooling layers, and a one-layer FC. The kernel numbers of the 3D convolution

layers were 20, 35, 35, and 35, respectively. This is a typical method that uses both spatial features and LSF for classification.

MCM [22]: He et al., constructed multiscale HSI cubes by increasing the size of the spatial windows of the center pixel. A covariance map was generated for each scale to represent the information of the central pixel, and the covariance maps obtained by various scales were used to generate multiscale covariance maps (MCM). MCMs were further used for classification using the same classification network proposed in this study. This is a typical method that uses spatial features and manually designed GSF for classification.

LGSF: The local-global spectral feature proposed in this study, which fully considers the LSF, GSF, and spatial features in the HSI. Moreover, the extraction and generation of LSF and GSF are both automatic and data-driven.

The following common metrics were used to evaluate the performance of the methods: producer's accuracy (for each class, PA), overall accuracy (OA), average accuracy (AA), and Kappa coefficient (KC). Moreover, considering that the small number of training samples and the random initialization of parameters may introduce uncertain effects and cause the classification accuracy to fluctuate, each method was evaluated three times, and the mean accuracy was considered the final accuracy score.

## 4. Results and Discussion

Each method listed above was trained and modeled until convergence, and the PA, OA, AA, and KC were used to measure their accuracies. The PA describes the accuracy of each class, and a higher PA indicates a more accurate result for the corresponding class. The OA describes the accuracy of all pixels, and a higher OA indicates that more pixels are classified correctly. However, OA can be affected by classes in large numbers. AA is the average PA of each class, and a higher AA indicates a better accuracy in all classes. KC is a comprehensive evaluation index, and a higher KC indicates not only a higher accuracy but also less misclassification of each class.

Tables 5–8 report the PA, OA, AA, and KC of all methods on the Houston 2013, Houston 2018, Pavia University, and Salinas Valley datasets, respectively. The highest accuracies are highlighted in bold. Overall, our proposed LGSF outperforms all comparison methods and has the highest OA, AA, and KC on the four datasets. The SVM has the lowest accuracy among the comparison methods. By combining the results of these methods with their characteristics, we observed that sufficiently describing the spectral features leads to a higher accuracy. Moreover, the comparison results show that introducing spatial information can promote the classification accuracy greatly. However, we also discovered that the spatial and spectral features are suitable for different classes, which means that spatial and spectral features should be selectively used according to class characteristics.

**Table 5.** Classification results of different methods on the Houston 2013 dataset. (shown in %).

| Class Name | SVM | 1-D CNN | R2D | 2-D CNN | 3-D CNN | MCM | LGSF |
|---|---|---|---|---|---|---|---|
| Healthy grass | 93.20 | 95.78 | 98.50 | 98.19 | 98.06 | 96.46 | **99.46** |
| Stressed grass | **99.09** | 98.08 | 98.53 | 98.31 | 94.01 | 48.58 | 98.03 |
| Synthetic grass | 21.78 | 68.08 | **97.53** | 28.13 | 45.35 | 94.57 | 58.57 |
| Trees | 98.13 | 80.04 | **98.76** | 98.63 | 98.47 | 67.88 | 87.45 |
| Soil | 85.73 | 92.91 | 91.88 | 97.56 | 95.42 | 99.03 | **99.74** |
| Water Soil | **100.00** | 14.28 | 22.98 | 46.50 | 45.96 | 86.10 | 65.58 |
| Residential | 52.40 | 62.99 | 65.37 | **89.09** | 73.85 | 40.35 | 80.11 |
| Commercial | 94.19 | 69.21 | 79.69 | 79.62 | 64.27 | 72.19 | **94.25** |
| Road | 44.98 | 70.25 | 83.66 | **93.51** | 79.72 | 88.92 | 85.86 |
| Highway | 58.95 | 66.28 | **87.56** | 74.04 | 70.17 | 73.22 | 85.81 |
| Railway | 45.31 | 59.11 | 81.43 | 75.00 | 60.97 | 26.42 | **86.15** |
| Parking Lot 1 | 8.53 | 71.25 | 90.06 | 77.71 | 68.59 | 77.95 | **92.15** |
| Parking Lot 2 | 19.27 | 18.76 | 41.62 | 77.86 | 44.89 | 67.47 | **82.15** |
| Tennis Court | 50.10 | 56.42 | 88.43 | 90.71 | 85.76 | 87.40 | **98.68** |
| Running Track | 99.36 | 98.17 | **99.93** | 92.86 | 97.74 | 88.12 | 93.52 |
| **OA** | 59.21 | 69.65 | 80.89 | 80.26 | 72.49 | 53.72 | **85.30** |
| **AA** | 64.73 | 68.11 | 81.73 | 81.18 | 74.88 | 74.31 | **87.17** |
| **KC** | 56.06 | 67.31 | 79.38 | 78.66 | 70.27 | 49.59 | **84.09** |

**Table 6.** Classification results of different methods on the Houston 2018 dataset. (shown in %).

| Class No. | SVM | 1-D CNN | R2D | 2-D CNN | 3-D CNN | MCM | LGSF |
|---|---|---|---|---|---|---|---|
| Healthy grass | 51.92 | 62.87 | **79.18** | 69.86 | 68.03 | 71.81 | 65.40 |
| Stressed grass | 60.44 | 69.80 | 89.56 | 91.68 | 86.17 | **92.37** | 91.58 |
| Artificial turf | 2.31 | 2.12 | 84.44 | 77.50 | 63.99 | **96.77** | 96.05 |
| Evergreen trees | 75.44 | 73.89 | 77.24 | 82.35 | **86.34** | 81.07 | 75.17 |
| Deciduous trees | 10.98 | 11.52 | 32.56 | 26.73 | 20.34 | 28.74 | **32.90** |
| Bare earth | 7.91 | 8.70 | 59.95 | 43.68 | 39.62 | 85.79 | **89.81** |
| Water | **100.00** | 7.42 | 25.90 | 20.92 | 12.54 | 39.97 | 46.41 |
| Residential buildings | 44.21 | 46.63 | 65.13 | 65.52 | 63.79 | 64.59 | **75.19** |
| Non-res. buildings | 98.31 | **98.75** | 96.55 | 96.90 | 95.96 | 97.29 | 98.04 |
| Roads | 27.29 | 18.64 | 52.58 | 59.05 | 44.62 | 56.71 | **59.36** |
| Sidewalks | 36.47 | 30.00 | 47.61 | **51.01** | 47.64 | 50.45 | 43.56 |
| Crosswalks | 1.58 | 1.89 | 4.78 | 5.14 | 4.15 | 6.47 | **6.67** |
| Major thoroughfares | 38.94 | 28.01 | 61.35 | 65.96 | 58.51 | 70.09 | **78.40** |
| Highways | 13.08 | 15.50 | 38.55 | 54.25 | 39.80 | 50.56 | **61.75** |
| Railways | 13.97 | 14.28 | 84.14 | 65.40 | 68.21 | 88.67 | **95.80** |
| Paved parking lots | 9.92 | 10.16 | 69.43 | 55.51 | 42.51 | 72.08 | **82.19** |
| Unpaved parking lots | 0.09 | 0.29 | 8.99 | 2.59 | 3.35 | 6.54 | **16.65** |
| Cars | 11.45 | 9.95 | 27.09 | 34.11 | 32.85 | 36.39 | **64.46** |
| Trains | 17.64 | 10.17 | 24.24 | 59.69 | 48.05 | 60.27 | **76.21** |
| Stadium seats | 35.89 | 24.42 | 50.28 | **57.32** | 55.88 | 55.51 | 49.23 |
| **OA** | 24.05 | 31.94 | 66.98 | 69.89 | 63.79 | 72.75 | **76.09** |
| **AA** | 32.89 | 27.25 | 53.98 | 54.26 | 49.12 | 60.61 | **65.24** |
| **KC** | 21.22 | 26.69 | 60.30 | 63.60 | 56.79 | 66.55 | **70.41** |

**Table 7.** Classification results of different methods on the Pavia University dataset. (shown in %).

| Class No. | SVM | 1-D CNN | R2D | 2-D CNN | 3-D CNN | MCM | LGSF |
|---|---|---|---|---|---|---|---|
| Asphalt | 96.40 | 93.38 | 96.92 | 98.78 | 97.60 | 99.25 | **99.67** |
| Meadows | 89.94 | 86.68 | 96.88 | 99.20 | 95.66 | 99.78 | **99.81** |
| Gravel | 39.36 | 36.98 | 69.44 | 82.97 | 81.25 | 94.75 | **98.44** |
| Trees | 53.33 | 50.82 | 90.28 | 98.54 | **98.83** | 97.63 | 96.69 |
| Painted metal sheets | 98.41 | 98.33 | 99.52 | 99.28 | **100.00** | 99.92 | 99.36 |
| Bare Soil | 37.03 | 38.28 | 79.46 | 97.00 | 72.41 | 99.37 | **99.59** |
| Bitumen | 34.88 | 39.72 | 70.72 | 76.57 | 77.48 | 87.24 | **95.71** |
| Self-Blocking Bricks | 66.54 | 71.42 | 81.83 | 91.35 | 87.67 | 90.39 | **96.18** |
| Shadows | 99.88 | 98.32 | 99.65 | 99.49 | **99.88** | 99.72 | 96.10 |
| **OA** | 64.86 | 65.96 | 90.38 | 96.43 | 91.02 | 97.97 | **98.94** |
| **AA** | 68.42 | 68.21 | 87.19 | 93.69 | 90.09 | 96.45 | **97.95** |
| **KC** | 57.01 | 57.81 | 87.29 | 95.25 | 88.13 | 97.29 | **98.59** |

**Table 8.** Classification results of different methods on the Salinas Valley dataset. (shown in %).

| Class No. | SVM | 1-D CNN | R2D | 2-D CNN | 3-D CNN | MCM | LGSF |
|---|---|---|---|---|---|---|---|
| Brocoli_green_weeds_1 | 91.99 | 97.71 | **100.00** | **100.00** | **100.00** | 99.96 | **100.00** |
| Brocoli_green_weeds_2 | 99.10 | 99.51 | 99.88 | **100.00** | 99.78 | 99.80 | 99.93 |
| Fallow | 84.64 | 93.60 | 99.24 | 99.52 | 97.48 | 99.12 | **99.95** |
| Fallow_rough_plow | 97.70 | 98.51 | 98.37 | 99.64 | **100.00** | 98.24 | 98.91 |
| Fallow_smooth | 97.69 | 98.63 | 98.98 | **99.94** | 99.15 | 99.38 | 99.74 |
| Stubble | **100.00** | **100.00** | 99.92 | 99.97 | 99.17 | 99.97 | 99.95 |
| Celery | 95.76 | 96.94 | 99.86 | 99.87 | **99.92** | 99.26 | 99.88 |
| Grapes_untrained | 71.33 | 78.15 | 85.54 | 90.89 | 90.75 | 95.57 | **97.49** |
| Soil_vinyard_develop | 96.66 | 96.58 | 98.85 | 99.95 | 99.85 | 99.81 | **99.95** |
| Corn_senesced_green_weeds | 84.56 | 85.12 | 94.54 | 98.19 | 91.94 | 95.76 | **98.35** |

**Table 8.** *Cont.*

| Class No. | SVM | 1-D CNN | R2D | 2-D CNN | 3-D CNN | MCM | LGSF |
|---|---|---|---|---|---|---|---|
| Lettuce_romaine_4wk | 83.41 | 76.77 | 98.16 | 98.26 | 97.61 | 99.42 | **99.76** |
| Lettuce_romaine_5wk | 97.15 | 96.17 | 97.19 | 99.98 | 99.78 | 99.29 | **100.00** |
| Lettuce_romaine_6wk | 92.69 | 93.55 | 98.40 | **100.00** | 99.84 | 98.55 | 99.88 |
| Lettuce_romaine_7wk | 94.43 | 93.45 | 91.98 | **99.86** | 97.44 | 98.90 | 99.79 |
| Vinyard_untrained | 60.86 | 66.22 | 71.12 | 79.61 | 82.77 | 88.47 | **92.44** |
| Vinyard_vertical_trellis | 98.21 | 93.40 | 95.79 | 99.71 | 99.49 | 99.47 | **100.00** |
| **OA** | 85.44 | 87.88 | 91.77 | 94.95 | 94.81 | 96.88 | **98.23** |
| **AA** | 90.39 | 91.52 | 95.49 | 97.84 | 97.18 | 98.19 | **99.13** |
| **KC** | 83.74 | 86.48 | 90.83 | 94.37 | 94.21 | 96.52 | **98.03** |

### 4.1. Comparison of Different Datasets

For the Houston 2013 dataset, the OA, AA, and KC of our LGSF were 85.30%, 87.17%, and 84.09%, respectively (Table 5). Compared with all baselines, LGSF showed great advantages in terms of overall statistical metrics. There were seven classes in which LGSF surpassed the other methods. Most classes in this dataset achieved 80% accuracy using LGSF, except for synthetic grass and water soil. The highest accuracy of these two classes was obtained using R2D and SVM, which are both methods based only on spectral information. Houston 2013 is a city dataset whose classes cover a wide range, and the spectral properties of classes vary greatly. Thus, this dataset was suitable for testing the applicability of these methods. Classification methods have poor applicability when they show high accuracy on a few classes, but low accuracy on other classes, such as SVM. Our LGSF surpassed the other methods on seven classes and has the highest AA, which reveals its effectiveness.

For the Houston 2018 dataset, the OA, AA, and KC of our LGSF were 76.09%, 65.24%, and 70.41%, respectively (Table 6), which were the highest among the evaluated methods. There were 12 classes in which LGSF surpassed the other methods, which was also the highest among all methods. In the Houston 2018 dataset, the accuracy fluctuated greatly among different methods, with a 50% difference between the highest and lowest accuracy, and there was one class in which each method had an accuracy of less than 10%. The reason may be that this dataset is relatively complex owing to its numerous classes and small band number, which is insufficient spectral information for distinguishing targets. The superiority of our method on such datasets verifies the effectiveness of the proposed method for spectral utilization.

For the Pavia University dataset, the OA, AA, and KC of our LGSF were 98.94%, 97.95%, and 98.59%, respectively (Table 7), which were the highest among the evaluated methods. There were six classes in which LGSF surpassed the other methods and the accuracies of the remaining classes were very close to the highest accuracy. The LGSF accuracy on each class was greater than 95%, which shows great advantages compared with other methods. The accuracy for this dataset was much higher than that for the Houston 2013 and Houston 2018 datasets, which are under the city scenario. The reason may be that the spatial coverage and category numbers of this dataset are both smaller than those of the above two datasets.

For the Salinas Valley dataset, the OA, AA, and KC of our LGSF were 98.23%, 99.13%, and 98.03%, respectively (Table 8). In this dataset, a few classes achieved 100% accuracy using various methods. LGSF had the best accuracy for nine classes, and the accuracies of most class were close to 99%. This may be because this dataset is a vegetation dataset, which is relatively simple compared to the above three datasets under the complex city scenario. Moreover, sufficient spectral information in this dataset (containing most bands among the four datasets) is also helpful for classification.

### 4.2. Comparison of Different Methods

Comparing the performances of methods that only use spectral information (i.e., SVM, 1D CNN, and R2D), we saw that the accuracy increases significantly with the refinement of the design of spectral features (SVM with no LSF, 1D CNN with LSF, and R2D with optimized LSF) on all datasets, which emphasizes the importance of LSF as well as the effectiveness of transforming the 1D spectral vector into a 2D spectral image.

Comparing the performances of methods that contain spatial information (i.e., 2D CNN, 3D CNN, MCM, and LGSF), it is surprising that 2D CNN has a better accuracy than 3D CNN on all datasets, which implies that simply combining spatial and spectral information using 3D convolution may not be optimal. We noticed that the difference in network structure (e.g., layer number and feature number) can also lead to this situation; however, it also means that 3D CNN requires a more refined design than 2D CNN.

Comparing the performance of methods with and without spatial information, we observed that there is considerable salt and pepper noise in the classification results of methods with no spatial information (Figure 7c–e, Figure 8c–e, Figures 9c–e and 10c–e), whereas the classification results of methods with spatial information (Figure 7f–i, Figure 8f–i, Figures 9f–i and 10f–i) are smoother. This demonstrates that introducing spatial information can improve robustness.

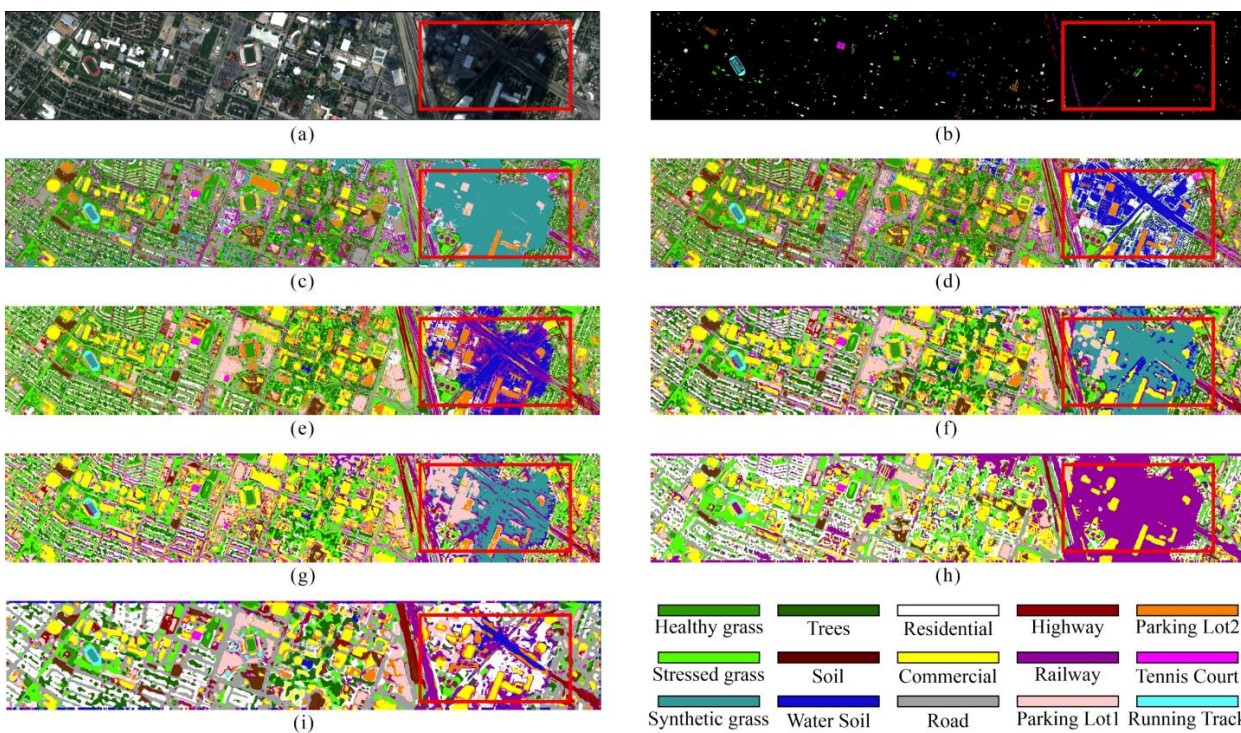

**Figure 7.** The classification results obtained using different methods on the Houston 2013 dataset. (**a**) Three-band composite image of the Houston 2013 HSI. (**b**) The ground truth of Houston 2013 HSI. The classification results of (**c**) SVM, (**d**) 1D CNN, (**e**) R2D, (**f**) 2D CNN, (**g**) 3D CNN, (**h**) MCM, and (**i**) LGSF.

Comparing the performance of the LGSF with other methods, we observed that the MCM and LGSF exhibited the top two accuracies on almost all datasets. The common point between them is the use of GSF, highlighting its importance. LGSF surpasses MCM on all datasets, and MCM has an abnormal drop in the Houston 2013 dataset, which may be due to the lack of LSF or the limitation of manually designed GSF in MCM. It can be concluded that with the combination of LSF, GSF, and spatial information, our LGSF can obtain better accuracy of HSI classification. Moreover, in the Houston 2013 dataset, there was a large dark area caused by clouds (marked with a red box, where most classes were related to

urban scenes). The spectral information of ground objects is strongly changed in this area, and most methods misclassified this area into classes that appear dark in the image, such as water (Figure 7d,e) or synthetic grass (Figure 7c,f,g). However, LGSF (Figure 7i) classified this area into residential and parking lots, which are close to the original classes (all belong to the urban scene). This may be because the GSF in LSGF describes the relative shape of the spectral curve, which shows a certain similarity between the urban classes. This also means that LGSF has a better spectrum understanding ability. A similar situation also appears in the stadium seat class (marked with a red box) in the Houston 2018 dataset. Most comparison methods misclassified these pixels into classes with similar artificial materials, such as cars and trains (Figure 8c,e–g), or even water (Figure 8d,h), because of the existence of shadows. Our LGSF (Figure 8i) retained a good classification ability and classified this area with high accuracy.

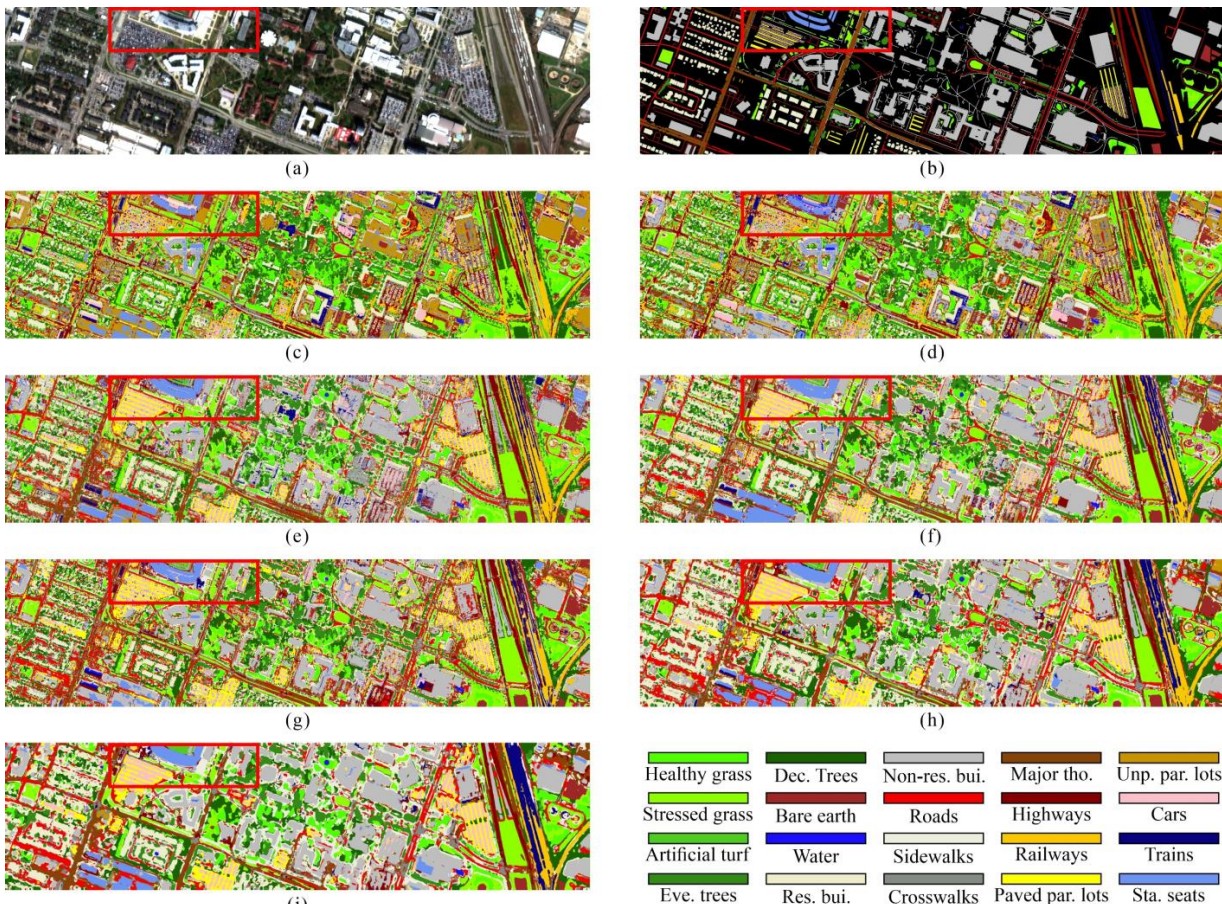

**Figure 8.** The classification results obtained using different methods on the Houston 2018 dataset. (**a**) Three-band composite image of the Houston 2018 HSI. (**b**) The ground truth of Houston 2018 HSI. The classification results of (**c**) SVM, (**d**) 1D CNN, (**e**) R2D, (**f**) 2D CNN, (**g**) 3D CNN, (**h**) MCM, and (**i**) LGSF.

### 4.3. Comparison of Different Land Use Classes

The above datasets can be roughly divided into four common land use classes: artificial targets, vegetation, water, and bare soil.

For the artificial targets, most road classes (e.g., road and highway in Houston 2013, and roads, sidewalks, crosswalks, and highways in Houston 2018) are associated with higher accuracy when using methods that contain spatial information (e.g., 2D CNN) than those methods that rely only on spectral information (e.g., SVM, 1D CNN, and R2D). This may be due to the strong spectral similarities of these artificial roads, while their spatial

texture features are more distinguishable than the spectral features. Thus, for road classes, the use of spatial features is more beneficial than spectral features. The same principles apply to residential classes whose spectral information can be messy and whose spatial information is more representative. However, for other artificial targets made of unique materials (e.g., the railway in Houston 2013 and Houston 2018), spectral-based methods have better classification capabilities than spatial-based methods.

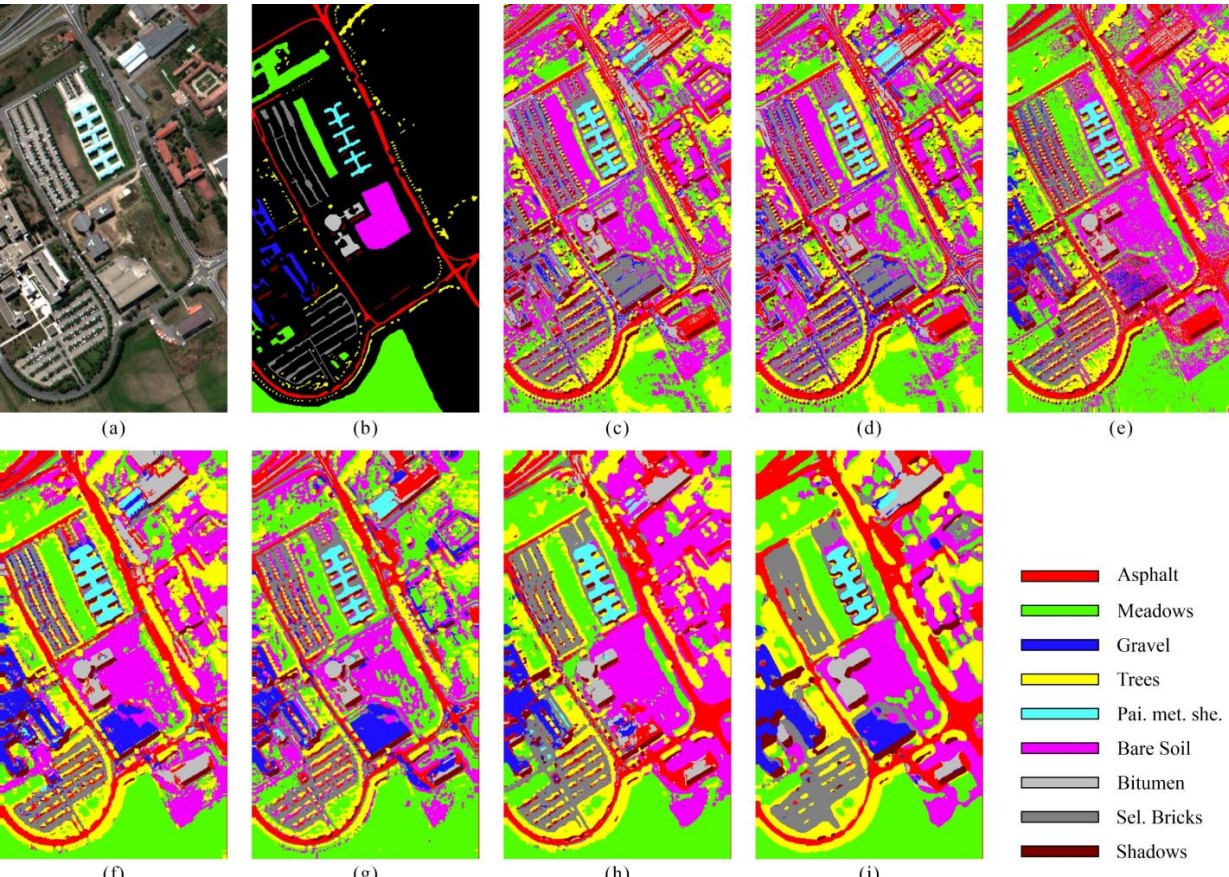

**Figure 9.** The classification results obtained using different methods on the Pavia University dataset. (**a**) Three-band composite image of the Pavia University HSI. (**b**) The ground truth of Pavia University HSI. The classification results of (**c**) SVM, (**d**) 1D CNN, (**e**) R2D, (**f**) 2D CNN, (**g**) 3D CNN, (**h**) MCM, and (**i**) LGSF.

For vegetation, mainly in the Salinas Valley and Pavia University datasets, we discovered that the difference in accuracy between methods is much smaller than that of other classes, which indicates that the classification accuracy of vegetation will not be too poor using any of the methods to classify the vegetation. However, under the demands of refined vegetation classification, it is still important to design appropriate spectral features to reflect the characteristics of different vegetation types. For example, the Lettuce_romaine_wk series in the Salinas Valley dataset has an accuracy of over 90% for most methods. However, methods that use LSF or GSF have more stable and accurate results.

For water, SVM obtained the highest accuracy in Houston 2013 (e.g., water soil) and Houston 2018 (e.g., water). However, the OA, AA, and KC of SVM were the lowest for both datasets. This is an interesting phenomenon, and it indicates that the water class has completely different characteristics from those of the other classes. MCM and LGSF obtained the second highest accuracy for the water class in Houston 2013 and Houston 2018, respectively, both of which contain GSF. Considering that SVM uses the original spectral curve and the GSF is a description of the overall shape of the spectral curve, we believe

that more global spectral information (complete spectral information or GSF) should be used to classify water, rather than the local spectral information.

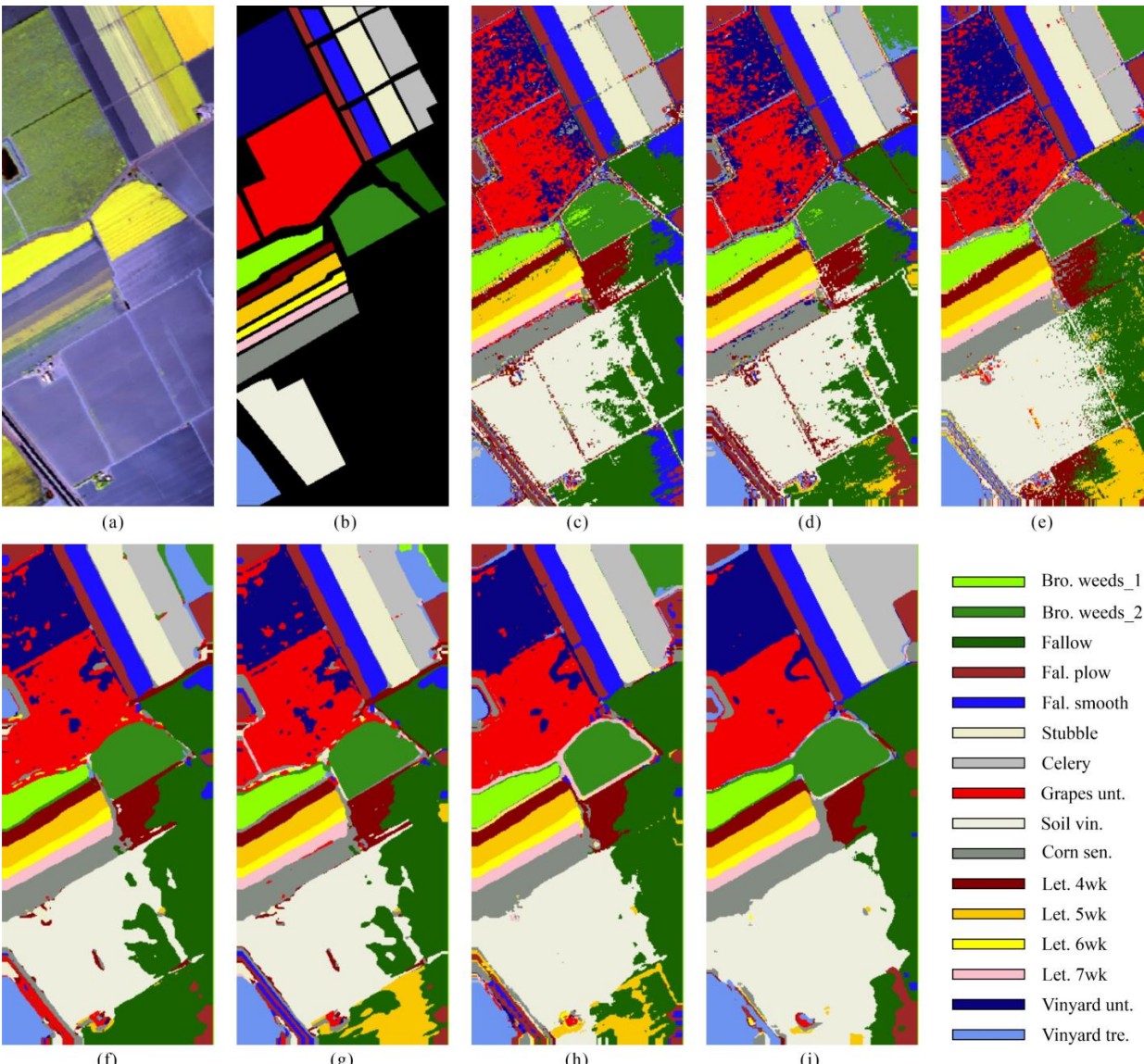

**Figure 10.** The classification results obtained using different methods on the Salinas Valley dataset. (**a**) Three-band composite image of the Salinas Valley HSI. (**b**) The ground truth of Salinas Valley HSI. The classification results of (**c**) SVM, (**d**) 1D CNN, (**e**) R2D, (**f**) 2D CNN, (**g**) 3D CNN, (**h**) MCM, and (**i**) LGSF.

For bare soil (the soil in the Houston 2013, the bare earth in the Houston 2018, and bare soil in the Pavia University dataset), MCM and LGSF obtained the best results. In particular, the accuracy of MCM was the lowest in the Houston 2013 dataset; however, its accuracy for soil was over 99%. Both MCM and LGSF contain GSF, which means that GSF is a necessary feature for identifying bare land. Moreover, we also noticed that introducing spatial information is helpful for improving the accuracy of bare soil, as the accuracies of 2D CNN and 3D CNN were higher than those of SVM, 1D CNN, and R2D. This may be because the heterogeneity of bare soil is relatively high, and there are numerous pixels belonging to other classes in the bare soil, such as grass, which confuse classifiers. The above analysis shows that the classification of bare soil should rely on spatial information as well as GSF.

### 4.4. Comparison of Ablation Experiments

To further demonstrate the effectiveness and contribution of each component in this paper, a series of ablation experiments were conducted, details shown in Table 9. The AE_LGSF contained the complete components of this paper. Based on AE_LGSF, the AE_2D removed the 2D transformation and used the original 1D spectral vector; the AE_LSF removed the LSFEM; the AE_GSF removed the GSFEM; the AE_SFOL removed the SFOL and used only cross-entropy loss for classification; the AE_DCBN used the traditional convolution-based network instead of the dilated convolution-based network (DCBN) for classification with same number of feature maps in each layer; the AE_SPAT removed the spatial enhancement. By comparing each ablation experiment that removed a specific component with the AE_LGSF, the contribution of the removed component to the classification results can be verified. The results of these ablation experiments on four datasets are shown in Tables 10–13. It can be seen from the tables, when any component is removed, that the classification accuracy decreases to a certain extent. In the following, we discuss the results from three perspectives (data input, feature extraction and feature classification), which cover the entire process of HSI classification.

**Table 9.** The design of ablation experiments.

|          | Trans_2D | LSFEM | GSFEM | SFOL | DCBN | Spatial Enhance |
|----------|:--------:|:-----:|:-----:|:----:|:----:|:---------------:|
| AE_LGSF  | √        | √     | √     | √    | √    | √               |
| AE_2D    | ×        | √     | √     | √    | √    | √               |
| AE_LSF   | √        | ×     | √     | √    | √    | √               |
| AE_GSF   | √        | √     | ×     | √    | √    | √               |
| AE_SFOL  | √        | √     | √     | ×    | √    | √               |
| AE_DCBN  | √        | √     | √     | √    | ×    | √               |
| AE_SPAT  | √        | √     | √     | √    | √    | ×               |

**Table 10.** Classification results of ablation experiments on the Houston 2013 dataset. (shown in %).

|     | AE_2D | AE_LSF | AE_GSF | AE_SFOL | AE_DCBN | AE_SPAT | AE_LGSF |
|-----|-------|--------|--------|---------|---------|---------|---------|
| OA  | 85.06 | 83.41  | 82.81  | 84.67   | 82.82   | 80.95   | 85.30   |
| AA  | 83.37 | 87.13  | 84.72  | 87.00   | 83.21   | 82.88   | 87.17   |
| KC  | 83.87 | 81.99  | 81.34  | 83.41   | 81.40   | 79.36   | 84.09   |

**Table 11.** Classification results of ablation experiments on the Houston 2018 dataset. (shown in %).

|     | AE_2D | AE_LSF | AE_GSF | AE_SFOL | AE_DCBN | AE_SPAT | AE_LGSF |
|-----|-------|--------|--------|---------|---------|---------|---------|
| OA  | 75.82 | 75.55  | 74.53  | 74.83   | 75.11   | 67.80   | 76.09   |
| AA  | 61.62 | 62.02  | 61.86  | 65.09   | 62.55   | 55.71   | 65.24   |
| KC  | 70.20 | 69.87  | 68.65  | 69.03   | 69.07   | 61.01   | 70.41   |

**Table 12.** Classification results of ablation experiments on the Pavia University dataset. (shown in %).

|     | AE_2D | AE_LSF | AE_GSF | AE_SFOL | AE_DCBN | AE_SPAT | AE_LGSF |
|-----|-------|--------|--------|---------|---------|---------|---------|
| OA  | 98.58 | 98.16  | 98.44  | 98.77   | 98.59   | 89.91   | 98.94   |
| AA  | 97.61 | 96.84  | 97.38  | 97.93   | 97.98   | 87.06   | 97.95   |
| KC  | 98.11 | 97.54  | 97.92  | 98.36   | 98.12   | 86.64   | 98.59   |

**Table 13.** Classification results of ablation experiments on the Salinas Valley dataset. (shown in %).

|     | AE_2D | AE_LSF | AE_GSF | AE_SFOL | AE_DCBN | AE_SPAT | AE_LGSF |
|-----|-------|--------|--------|---------|---------|---------|---------|
| OA  | 98.05 | 97.97  | 97.24  | 97.79   | 97.24   | 91.83   | 98.23   |
| AA  | 98.97 | 98.76  | 98.76  | 98.86   | 98.76   | 95.26   | 99.13   |
| KC  | 97.83 | 97.74  | 96.92  | 97.53   | 96.92   | 90.90   | 98.03   |

For the data input part, the directly related experiment is the AE_SPAT. There is a large decrease in accuracy of AE_SPAT, which indicates the importance of spatial enhancement. The reason is that the extraction of spectral features is the statistical induction based on stable spectral patterns. However, the ubiquitous noise information in HSI makes the spectral value of a single pixel fluctuate within a certain range, resulting in a lack of stable characterization. Therefore, by introducing the spatial constraints and establishing a relationship between the spectra of surrounding pixels and central pixels, the effect of noise can be reduced and the statistical nature of the spectrum can be enhanced, thereby extracting more stable features and improving classification accuracy. Thus, it is important to using the spatial information to enhance and stabilize spectral information before feature extraction in HSI classification.

For the feature extraction part, the directly related experiments are the AE_2D, AE_LSF, AE_GSF and AE_SFOL. (1) The decrease of AE_2D reveals the positive effect of 2D transformation on spectral feature extraction, which can be explained in two perspectives. From a row perspective, in the 2D spectral image, the LSF that extracted by 1D convolution are still preserved in the 2D receptive field. Moreover, the relationship between LSF in different rows can be further mined and combined, which means that the 2D pattern can obtain more diverse spectral features without losing the LSF obtained by 1D pattern. From a column perspective, in the 2D spectral image, the band combinations of columns are regular, and express another LSF (i.e., LSF of uniformly spaced $K$ bands, where $K$ is the width of 2D image). This is similar with the idea of dilated convolution that enlarge the receptive fields with the dilated rate. In this way, the 2D pattern is conducive to capturing a wider range of LSF, and produces more diverse spectral features. (2) Comparing with the AE_LSF and AE_GSF (using the GSF and LSF for classification respectively), the AE_LSF outperformances the AE_GSF in three datasets, which indicates that the GSF may have a greater impact on classification than LSF. Therefore, when using CNN for HSI classification, it is necessary to jump out of the local receptive field of convolutional kernels and effectively design long-distance or distance-independent global features. (3) The decrease of AE_SFOL demonstrates the contribution of SFOL, which guides the network to automatically optimize LGSF for better category separability.

For the feature classification part, the directly related experiment is the AE_DCBN. The decrease of AE_DCBN indicates that based on good features, it is also important to use a good classifier for analysis and classification.

### 4.5. The Analysis and Discussion of Local and Global Characteristics of LGSF

Figure 11 shows the Pavia University dataset as an example of the extracted LGSF with the original spectral curve of each class. In LGSF maps, the pixels in each row and column represent the relationship between the spectra of the corresponding bands. For example, the pixel in rows 1 and columns 2 represents the relationship between bands 1 and 2. The redder areas indicate high values and can be roughly considered as activated areas, which contain more important features.

Analyzing the LGSF maps from a local perspective, we found that a larger local variation in the spectral curve causes a higher activation value. In Figure 11, the asphalt and gravel classes (Figure 11a,c) show activation in the LGSF maps where the corresponding reflectivity in the spectral curves fluctuate, while the shadow class (Figure 11i) activates the LGSF map on the left side of the spectral curve, where the reflectivity tends to decrease. This indicates that the local rapidly changing spectra in the spectral curve are more capable of representing the characteristics of the targets. This is consistent with the experience because the rapid change in a certain local part is generally caused by the unique nature of the targets. For example, for vegetation, the most typical feature is the rapid change in reflectivity between the red and near-infrared bands, which is also the basis of normalized difference vegetation index (NDVI). Bitumen is an interesting class (Figure 11g), whose reflectivity is generally flat, with only a slight jitter at the front and end of the curve. Even

so, our method can effectively extract spectral features to represent the characteristics of the spectral curve.

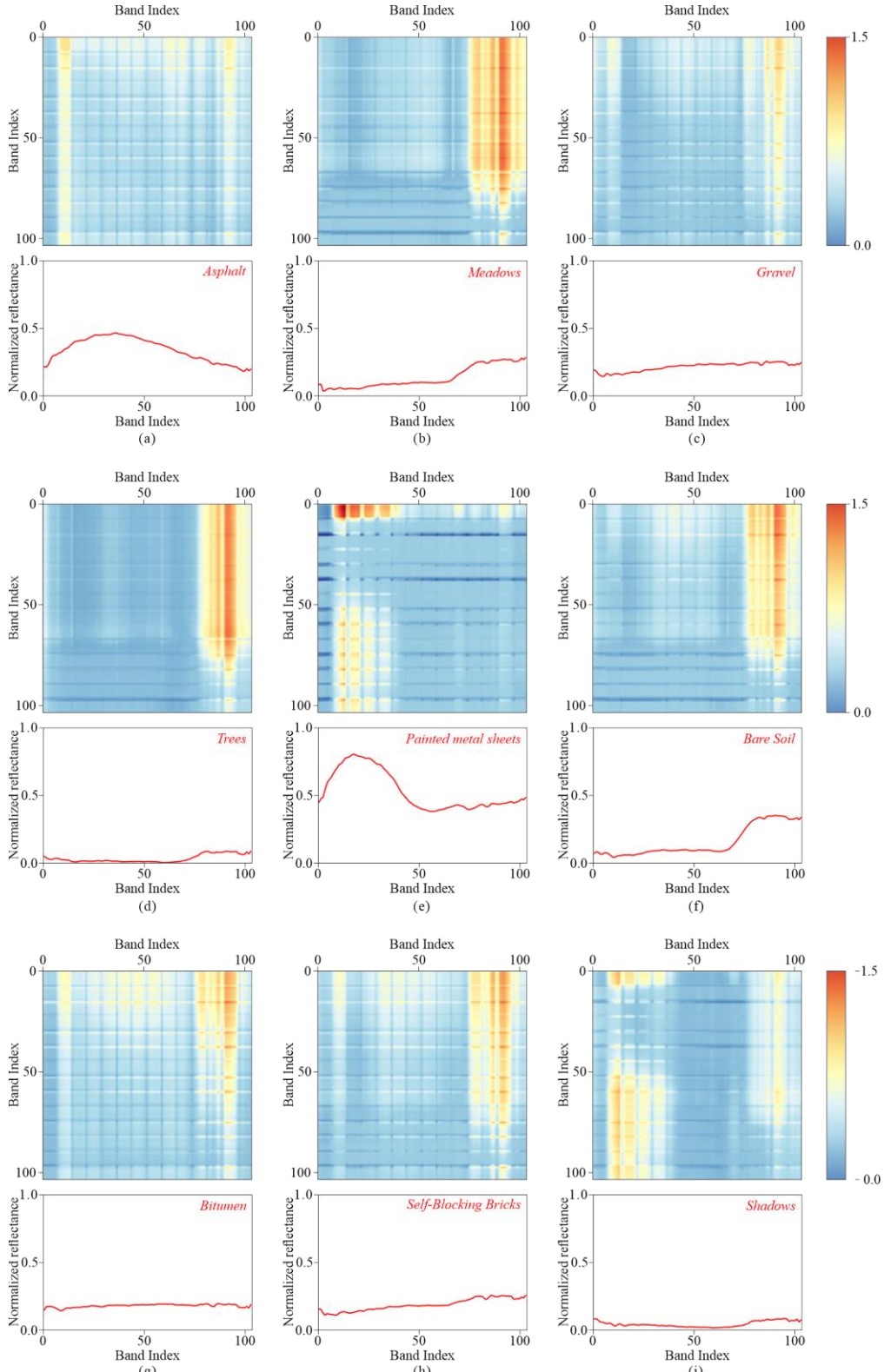

**Figure 11.** The LGSF and spectral curves of different classes from the Pavia University dataset.

Analyzing the LGSF maps from a global perspective, we found that the larger global variation in the spectral curve has a higher activation value, which is most obvious in the painted metal sheet class (Figure 11e). The spectral range of pixels with large activation values (on the left of the LGSF maps) corresponds to the reflection peak on the spectral curve. The spectra of the reflection peaks are greatly activated when combined with all other spectra, except for their adjacent spectra, because their values are similar to the surrounding spectra and differ greatly from others. The same phenomenon can also be observed in the meadows, trees, and bare soil classes (Figure 11b,d,f). The self-blocking bricks (Figure 11h) are a special class, whose reflectivity gradually increases with an increase in wavelength, which means that the farther the distance of the wavelength, the greater the difference in reflectivity. Therefore, in the LGSF maps, the activation value of the spectra with the farthest distance (the front and last bands) is the largest (the activation area on the top right), and the value gradually decreases with the decrease in spectral distance (the value gradually decreases from top to bottom and from right to left).

## 5. Conclusions

HSI contains rich spectral information and is widely used in a series of classification applications. However, the rich spectrum contained in HSI increases the difficulty of extracting useful hidden information for classification. The spectral features are considered to represent useful information in the spectrum as well as the basis of HSI classification. In this study, we summarize spectral features into two categories: LSF and GSF. The LSF describes the statistical information of the local and adjacent areas of the spectral curve, whereas the GSF describes the relative relationship between the long-distance and non-adjacent areas of the spectral curve. We demonstrated the importance of LSF and GSF when dealing with HSI and proposed a LGSF extraction and optimization method to extract and combine both. We first transformed the 1D spectral vector into 2D spectral images to increase the adjacency opportunities between spectra as well as increase the possibility of obtaining features with more forms. Next, the LSF was extracted using LSFEM, and the GSF was extracted using GSFEM upon the LSF to form the LGSF. The LGSF was optimized using the SFOL to maximize the class separability and was further enhanced with a spatial relation. A dilated convolution-based network was designed to obtain multiscale image features of LGSF and was used for HSI classification. We evaluated our method on four HSI datasets and compared it with several other methods that focus on various features for HSI classification. The experimental results showed that the proposed method achieved the highest accuracy compared with other methods that use single or incomplete LSF and GSF, which demonstrates that spectral information can be more effectively described after the extraction, combination, and optimization processes of local and global spectral features proposed in this article. Moreover, it also reveals that effective, full, and comprehensive use of spectral information can improve the classification accuracy of HSI and is of great significance to HSI application.

**Author Contributions:** Conceptualization, Z.X. and C.S.; methodology, Z.X. and S.W.; software, Z.X.; writing—original draft preparation, Z.X. and C.S.; supervision, C.S. and X.Z. All authors have read and agreed to the published version of the manuscript.

**Funding:** This research was funded by the National Natural Science Foundation of China, grant number 42050103 and the National Key R&D Program of China, grant number 2018YFB0505002.

**Data Availability Statement:** The datasets used in this article are all publicly available. The Houston 2013 dataset was downloaded from https://hyperspectral.ee.uh.edu/?page_id=459, accessed on 11 June 2022. The Houston 2018 dataset was downloaded from https://hyperspectral.ee.uh.edu/?page_id=1075, accessed on 11 June 2022. The Pavia University and Salinas Valley datasets were downloaded from https://www.ehu.eus/ccwintco/index.php/Hyperspectral_Remote_Sensing_Scenes, accessed on 11 June 2022.

**Acknowledgments:** We thank the institutions that collected the hyperspectral image datasets. We also thank the anonymous reviewers for their constructive comments.

**Conflicts of Interest:** The authors declare no conflict of interest.

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
