# Peer review of "Local and Global Spectral Features for Hyperspectral Image Classification"

_remotesensing, doi:10.3390/rs15071803_

Round 1

Reviewer 1 Report

The accuracy of the classification of remote sensing of ground objects has been improved by using local and global spectral features. The method is novel and has certain practical value. In particular, the manuscript highlights the advantages of the proposed method by comparing it with different methods. However, the results lack comparison with the experimental results of others, so I suggest that the author include this part in the content.

Reviewer 2 Report

Hyperspectral image classification using deep learning methods is a hot topic in recent years. This study proposed a local-global spectral feature extraction and optimization method for hyperspectral image classification. The proposed method combined local and global spectral features and outperformed the best results on some public hyperspectral datasets compared with other methods. The article is well-organized and interesting. Some minor issues:

1. Paragraph 2. In most of the literature, there are two methods to reduce the dimension of hyperspectral data: feature extraction and band selection. The two methods have different characteristics. So it is confusing that you use band selection to describe the feature extraction. 

2. Lines 174 to 180. The most contribution of this study is the proposed method of combing the local and global spectral features.

3. Line 497. Why OA can be affected by classes in large numbers?

4. Line 676. The full name of NDVI should be given.

Reviewer 3 Report

In this work, a local-global spectral feature extraction and optimization method is proposed for the task of HSI classification. The main components and designs includes: (1) 1D spectral vector to 2D spectral image reshape, (2) a two branch structure for the LSF feature extraction; (3) GSF feature extractor upon LSF feature representation; (4) contrastive learning objective; (5) classification; (6) spatial enhancement upon window-based average. Experiments on a series of benchmark HSI datasets demonstrate the effectiveness of the design as a whole, e.g., higher OA, AA, and KC values compared with selected methods. However, it seems that there lacks adequate discussions and empirical evidence to illustrate and verify the designing principles of each component. For example,

  • The LSFEM module handles the 2D spectral image as input with 2D convolutions, which extract the spatial local feature representations. However, spectral channels in the corresponding receptive fields correspond to the non-adjacent spectral features, how can we expect a good local spectral representation extraction?

  • It seems that there lacks ablations discussing the effectiveness of LFS and GFS, respectively, e.g., what if we only employ the LFS, or only the GFS? or firstly employ the GFS and then LFS considering the 2D spectral feature map characteristic? If possible, please provide more illustrations and well-designed ablations. 

  • There lacks empirical evidence of the spatial enhancement, does it really work or bring the performance benefits? If possible, please provide more empirical evidence to support. 

  • Similar thing happens about the dilation conv design. Please verify this point accordingly. 

Round 2

Reviewer 3 Report

The manuscript has been improved with more illustrations and ablation studies. While some of my concerns are not satisfactorily solved, it is recommended to be presented in the current position. Though I still strongly suggest the authors to include more discussions and in-depth analysis toward the proposed modules in their future works.